# Crisis response during the COVID-19 pandemic: Changes in social contact and social participation of older Americans

Yun Zhang [1,2], Amber Luo [3], Luke Hou[4], Shanquan Chen[5], Wei Zhang[6], Andrew Schwartz[7], Sean A. P. Clouston [8]*

**1** Program in Public Health, Renaissance School of Medicine, Stony Brook University, Stony Brook, New York, United States of America, **2** Pediatric Cardiovascular Clinical Research Core, Pediatric Heart Center, Columbia University Vagelos College of Physicians and Surgeons, Morgan Stanley Children's Hospital of New York-Presbyterian, New York, New York, United States of America, **3** Massachusetts Institute of Technology, Cambridge, Massachusetts, United States of America, **4** Ward Melville High School, Stony Brook, New York, United States of America, **5** Department of Population Health, Faculty of Epidemiology and Population Health, London School of Hygiene and Tropical Medicine, London, United Kingdom, **6** College of Public Health and Human Sciences, Oregon State University, Corvallis, Oregon, United States of America, **7** Department of Computer Science, Stony Brook University, Stony Brook, New York, United States of America, **8** Program in Public Health and Department of Family, Population and Preventive Medicine, Renaissance School of Medicine, Stony Brook University, Stony Brook, New York, United States of America

* sean.clouston@stonybrookmedicine.edu

**Data Availability Statement:** Data cannot be shared publicly because of its third-party ownership. Data are available from www.nhats.org

## Abstract

### Objective

This study aimed to assess changes in social contact with family, friends and healthcare providers, as well as social participation in working, volunteering, religious services and other organized activities, among older adults during the COVID-19 pandemic while examining the role of pre-COVID sociodemographic characteristics or cognitive and physical limitations in changes in social contact and participation.

### Methods

We conducted secondary data analyses in the National Health and Aging Trends Study (NHATS) COVID-19 questionnaire, collected in 2020 during a period of workplace closures and social distancing guidelines. We linked data to pre-COVID sociodemographic and medical information collected in 2019 before COVID interrupted social life. The frequency of participants' social contact and social participation prior to and during the COVID-19 pandemic, were compared using paired t-tests for summed scores. Multivariate linear regression was used to relate participants' socio-demographic, prior physical and cognitive performance with levels of social contact and participation and with changes reported during the COVID-19 pandemic.

### Results

In total, results from 2,486 eligible participants revealed that COVID-19 was associated with decreased social contact among family and friends (change: -0.62; SE: 0.06; p<0.0001),

for researchers who meet the criteria for access to confidential data.

**Funding:** Funding for this analysis was provided by the National Institute on Aging (NIH/NIA RF1AG058595). The National Health and Aging Trends Study (NHATS) is sponsored by the National Institute on Aging (grant number NIA U01AG032947) through a cooperative agreement with the Johns Hopkins Bloomberg School of Public Health. The funders had no role in study design, data collection and analysis, decision to publish, or preparation of the manuscript.

**Competing interests:** The authors have declared that no competing interests exist.

and social participation among older adults (change: -0.58; SE: 0.02; p<0.0001). Pre-COVID characteristics including older age, lower educational attainment, poorer physical performance, and more chronic conditions were associated with lower social contact and social participation and with how older adults adapted their social lives during the COVID-19 pandemic.

## Discussion

These results emphasize the importance of increasing digital inclusion for older adults in a major crisis.

## Introduction

The United States (US) is facing an emerging aging crisis, with the number of older American citizens expected to be 70 million by the year 2030 [1]. Age-related physical and cognitive decline among older adults have been linked to decreased autonomy [2], increased social isolation [3], and mortality. The growing proportion of older adults is predicted to substantially increase the burden on healthcare facilities, and nursing homes [1], and to increase the social and financial burden on families [4]. In older adults, social participation (e.g., active involvement in religious, sports, cultural, recreational, political, and volunteer community activities [3]) and social contact (e.g., the presence of robust social interactions) can stimulate physical activity and cognitive function among older adults, delaying autonomy loss while protecting against cognitive impairment [5] and mortality [6].

The novel Coronavirus Disease 2019 (COVID-19) pandemic rapidly upset traditional means of social contact and social participation across the US, disproportionately impacting the physical and mental well-being and social lives of older Americans [7–9]. Since the arrival of COVID in January 2020, social distancing policies began emerging that prevented large gatherings closing many religious, recreational, and community-focused social events and moving others immediately online [10]. However, older adults are more likely than younger people to lack access to digital resources including computers and the internet and often lack digital literacy to facilitate this move online [11, 12], the transition of means for social interactions online might have restricted access to social contact and social participation for older populations [13] magnifying the chances for exclusion from both in-person contact and digital services [14, 15]. Thus, older populations, who composed the majority of COVID-19–associated fatalities [7], were also suffering from more restricted opportunities for social contact and social participation during the COVID-19 pandemic [16].

To date, research has focused on characterizing the subjective loneliness and isolation of older people during the COVID-19 pandemic [17, 18], but few studies have investigated how the COVID-19 pandemic has transformed social contact and social participation in older American populations, especially in residential and community settings [19]. Yet, studies also report that in-person hospital visits for non-COVID-19 patients dropped significantly during the COVID-19 pandemic [20], raising concerns about the health of older adults experiencing acute illnesses or those requiring regular health monitoring by trained clinicians. Furthermore, the COVID-19 pandemic exerted disproportionate social impacts on disadvantaged populations, such as educationally disadvantaged populations and populations with comorbidities [7–9]. Since prior work has established evidence that sociodemographic and health factors are associated with social contact and social participation [21], there is a need to investigate crucial

relationships between sociodemographic and health factors and changes social contact and social participation.

Our research aims to describe and contrast changes in social contact and social participation of older adults residing in the U.S. during the COVID-19 pandemic. The goals of this study were to investigate: 1) social contact with family, friends and healthcare providers, social participation in working, volunteering, religious services and other organized activities, and their changes during the COVID-19 pandemic among US older adults, and 2) the effects of health-related measures (physical and cognitive performance), and sociodemographic factors (age, sex, race/ethnicity, education and marital status) on reported changes in social contact and social participation. We aimed to provide a comprehensive view of how the COVID-19 pandemic has affected the social lives of America's older adults. The results of our study could highlight sociodemographic and health factors affecting crisis-induced changes in social contact and social participation, providing informative data that can help policymakers and medical professionals to successfully promote age-friendly and inclusive policies in crisis time, especially amongst vulnerable older populations.

## Methods

### Ethics statement

The National Health and Aging Trends Study (NHATS) is sponsored by the National Institute on Aging and is being conducted by Johns Hopkins University Bloomberg School of Public Health. Johns Hopkins Bloomberg School of Public Health Institutional Review Board approved this study, and written informed consent was obtained from all the involved participants or their proxy respondents. As a secondary analysis using NHATS data in the public domain and anonymized, this research did not fall within the regulatory definition of research involving human subjects and thus was exempt from institutional board review. As a compatible use with what the participants have agreed to in the original consent form, we did not collect another one.

### Data

The NHATS is a nationally representative longitudinal study of Medicare beneficiaries aged 65 years or older living in a community setting (www.nhats.org). NHATS started in 2011 and was complemented in 2015 [22]. During the COVID-19 pandemic in 2020, the NHATS study distributed NHATS COVID-19 Questionnaire to conduct a supplemental study on participants' experiences and behaviors throughout the pandemic [23]. NHATS participants who completed a sample interview in Round 10 received a self-administered COVID-19 questionnaire. Proxies who answered the Round 10 NHATS interview for NHATS participants received a proxy version of the questionnaire about their representatives' experiences (the Proxy Questionnaire). Questionnaires were mainly distributed by mail from June-October 2020, when COVID mitigation strategies including limitations on the size of social gatherings and mask mandates in many social spheres were still common for the general population and were being consistently recommended specifically for adults aged 65 and older as well as those who were believed to be at heightened risk. Using NHATS' longitudinal tracking identifiers, we linked NHATS COVID-19 Questionnaire responses to data collected in Round 9 to measure pre-COVID physical and cognitive impairments and to identify sociodemographic information.

### Social contact

Social contact with family, friends, and social contact with healthcare providers is measured by the frequency of making phone calls; using emails, texts, or social media messages (includes

Facebook messages); making video calls (including Zoom, FaceTime, and other online videos); and in-person visits, obtained from the NHATS COVID-19 questionnaire data. Frequency of social contact was assessed on a four-item five-point scale (5 = at least daily, 4 = a few times a week, 3 = about once a week, 2 = less than once a week, 1 = never), with a higher score indicating a higher frequency of the measured activity. Total social contact with family, friends, and total social contact with healthcare providers before and during the COVID-19 pandemic were separately summed across the scale thus range from 4 to 20.

## Social participation

To measure social participation in our study, we referred to the NHATS COVID-19 questionnaire data to assess respondents whether working for pay (or in a business that one owns); volunteering; attending religious services; and attending clubs, classes, or other organized activities. Frequency was assessed by a four-item dichotomous scale (2 = participated online or in person, 1 = did not participate). Total social participation was summed across the four tests before and during the COVID-19 pandemic, ranging from 4 to 8.

**Cognitive performance.** Three types of cognitive information were collected in NHATS and was used to identify cognitively impaired participants. Following standard NHATS dementia criteria, we classified the cognitive status of participants into probable dementia, possible dementia, and no dementia [24–26]. Probable dementia was identified by the presence of at least one of the following: (1) a self-reported or proxy-reported dementia diagnosis; (2) AD-8 score ≥ 2 indicating probable dementia on the AD8 Dementia Screening Interview [27]; or (3) a score ≤1.5 standard deviations below the mean in at least two cognitive domains. If neither the conditions for possible dementia nor for probable dementia were satisfied, respondents were classified as "no dementia".

**Physical performance.** Physical health of the participants was assessed by the Short Physical Performance Battery (SPPB) to assess balance, lower extremity strength, and overall functional capacity of older adults [28]. The SPPB consists of three standing balance tests (side by side, semi-tandem, and full-tandem) to measure the duration a stand can be maintained, 5 timed chair stands, and gait speed on a 3-meter course. Each of the three tests was scored on a scale from 0 to 4, with higher numbers representing a better function. The summed scores across the three tests, ranging from 0 to 12 and, were classified into three levels: low function (0–4), moderate function (5–8), and high function (9–12).

**Covariables.** Sociodemographic measures included age (5-year age groups), sex, marital status, educational level, and race/ethnicity. Age was treated as an interval variable (5-year increments) based on available data. The number of chronic conditions for each participant was calculated to summarize the burden of the following nine chronic conditions among the participants: heart attack, heart disease, high blood pressure, arthritis, osteoporosis, diabetes, lung disease, stroke, and cancer. Thus, the number of chronic conditions ranged from 0–9.

## Statistical analyses

We calculated averages of social contact with family and friends, social contact with healthcare providers and social participation of the participants prior to and during the COVID-19 pandemic. The individual levels of social contact in 2019, prior to the COVID-19 pandemic, was matched to results collected during the COVID-19 pandemic and differences were calculated. Changes in total social contact with family and friends, total social contact with healthcare providers, and total social participation were examined using paired t-tests. For individual items of social contact with family, friends, and social contact with healthcare providers, which were paired categorical variables, we used Wilcoxon signed-rank test to assess significance; for an

individual item of social participation, which were paired dichotomous variables, we used McNemar's chi-square tests. To correct for heteroscedasticity, we utilized multivariate linear regression models with Huber–White robust standard errors to separately investigate associations between sociodemographic and prior physical and cognitive performance with (1) total social contact and social participation prior to and during the COVID-19 pandemic and (2) changes in social contact and social participation during the COVID-19 pandemic. Total social contact and total social participation prior to and during the COVID-19 pandemic, and their changes, were treated as continuous variables. All the final models accounted for age group, sex, race/ethnicity, education, marital status, earlier cognitive status, earlier physical performance, and the number of chronic conditions. Two-sided p-values <0.05 were treated as statistically significant. Statistical analyses used SAS statistical software version 9.4 (SAS Inc., Cary, NC).

## Results

From 3,961 eligible participants, the early-release NHATS SP COVID-19 Beta file includes responses from 3,188 participants or proxies who completed at least half of the items on any COVID research component and returned it before November 1, 2020. Our study excluded participants who reported a confirmed COVID-19 diagnosis and symptoms (*n* = 288) and results from individuals whose questionnaires were completed by proxy adults (*n* = 414). Because it is difficult to distinguish the impacts of social distancing for the public and special social distancing for COVID-19 diagnosis and symptoms, and there are differences between self-reported and proxy-reported experiences. In total, N = 2,486 participants were retained in the final analysis.

Most participants were female (*n* = 1,447; 58.21%), non-Hispanic White (1,974; 80.37%), or married/living with a partner (1,283; 51.61%). The average age was 80.78 years. Most participants had moderate (947, 41.35%) to high (850, 37.12%) physical performance. Based on dementia classification protocols, individuals were classified as probable dementia (64; 2.57%), possible dementia (114; 4.59%), or no dementia (2,308; 92.84%). Table 1 displays participants' characteristics including sociodemographic factors (age group, sex, race/ethnicity, marital status, and highest educational attainment), prior physical performance based on SPPB (Short Physical Performance Battery) classification, number of chronic conditions, and dementia status.

We present the changes in both individual and summed measures of social contact and social participation in older US adults during the COVID-19 pandemic (Fig 1). The total social contact with friends and family decreased significantly during the COVID-19 pandemic (change: -0.62; SE: 0.06; p<0.0001). On the other hand, total social contact with healthcare providers remained relatively unchanged (change: -0.01; SE: 0.03; p = 0.378). While total social contact with friends and family significantly decreased, the use of video calls with friends and family increased considerably during the pandemic (mean change: 0.13; SE: 0.02; p < 0.0001). Much of the observed decrease in social contact with friends and family can be attributed to a large decrease in in-person visits (change: -0.56; SE: 0.03; p<0.0001). Similarly, in-person visits to healthcare providers decreased significantly (change: -0.31; SE: 0.01; p<0.0001), while the use of video calls increased (mean change: 0.18; SE: 0.01; p<0.0001). The total frequency of social participation amongst our study sample decreased significantly (mean change: -0.58; SE: 0.02; p<0.0001). All individual measures of social participation decreased during the COVID-19 pandemic, with participation in clubs, classes, or other organized activities decreasing the most (change: -0.24; SE: 0.01), followed by participation in religious services (change: -0.17; SE: 0.01; p<0.0001), volunteering (change: -0.12; SE: 0.01; p<0.0001), and working for pay (change: -0.03; SE: 0.01; p<0.0001).

**Table 1. Sample characteristics.**

| | Frequency | Percent | | | | |
|---|---|---|---|---|---|---|
| **Age group** | | | | | | |
| 70–74 years old | 448 | 18.02 | | | | |
| 75–79 years old | 797 | 32.06 | | | | |
| 80–84 years old | 607 | 24.42 | | | | |
| 85 years and older | 634 | 25.50 | | | | |
| **Sex** | | | | | | |
| Male | 1,039 | 41.79 | | | | |
| Female | 1,447 | 58.21 | | | | |
| **Race/ethnicity** | | | | | | |
| White, non-Hispanic | 1,974 | 80.37 | | | | |
| Black, non-Hispanic | 356 | 14.50 | | | | |
| Other, non-Hispanic | 47 | 1.91 | | | | |
| Hispanic | 79 | 3.22 | | | | |
| **Education** | | | | | | |
| Less than high school | 265 | 10.75 | | | | |
| High school or equivalent | 620 | 25.16 | | | | |
| Beyond high school but less than college | 730 | 29.63 | | | | |
| Bachelor and beyond | 849 | 34.46 | | | | |
| **Marital status** | | | | | | |
| Married/Living with a partner | 1283 | 51.61 | | | | |
| Separated/Divorced | 343 | 13.80 | | | | |
| Widowed | 787 | 31.66 | | | | |
| Never married | 73 | 2.94 | | | | |
| **Earlier cognitive status** | | | | | | |
| Probable dementia | 64 | 2.57 | | | | |
| Possible dementia | 114 | 4.59 | | | | |
| No dementia | 2,308 | 92.84 | | | | |
| **SPPB score (physical performance)** | | | | | | |
| Low | 493 | 21.53 | | | | |
| Medium | 947 | 41.35 | | | | |
| High | 850 | 37.12 | | | | |
| **Number of chronic conditions** | | | | | | |
| 0 | 129 | 5.19 | | | | |
| 1 | 410 | 16.49 | | | | |
| 2 | 675 | 27.15 | | | | |
| 3 | 622 | 25.02 | | | | |
| 4 | 431 | 17.34 | | | | |
| 5 | 167 | 6.72 | | | | |
| 6 | 49 | 1.97 | | | | |
| 7 | 2 | 0.08 | | | | |
| 8 | 1 | 0.04 | | | | |
| **Total** | 2,486 | 100.00 | | | | |
| | **N** | **Minimum** | **Maximum** | **Mean** | **SE of Mean** | **Median** |
| **Age (year)** | 2,486 | 70.83 | 101.83 | 80.78 | 0.12 | 79.92 |
| **Number of chronic conditions** | 2,486 | 0 | 8 | 2.62 | 0.03 | 3 |

**Notes**: Analytic sample includes NHATS COVID-19 Questionnaire respondents who were self-respondents free of COVID-19 symptoms and diagnosis.

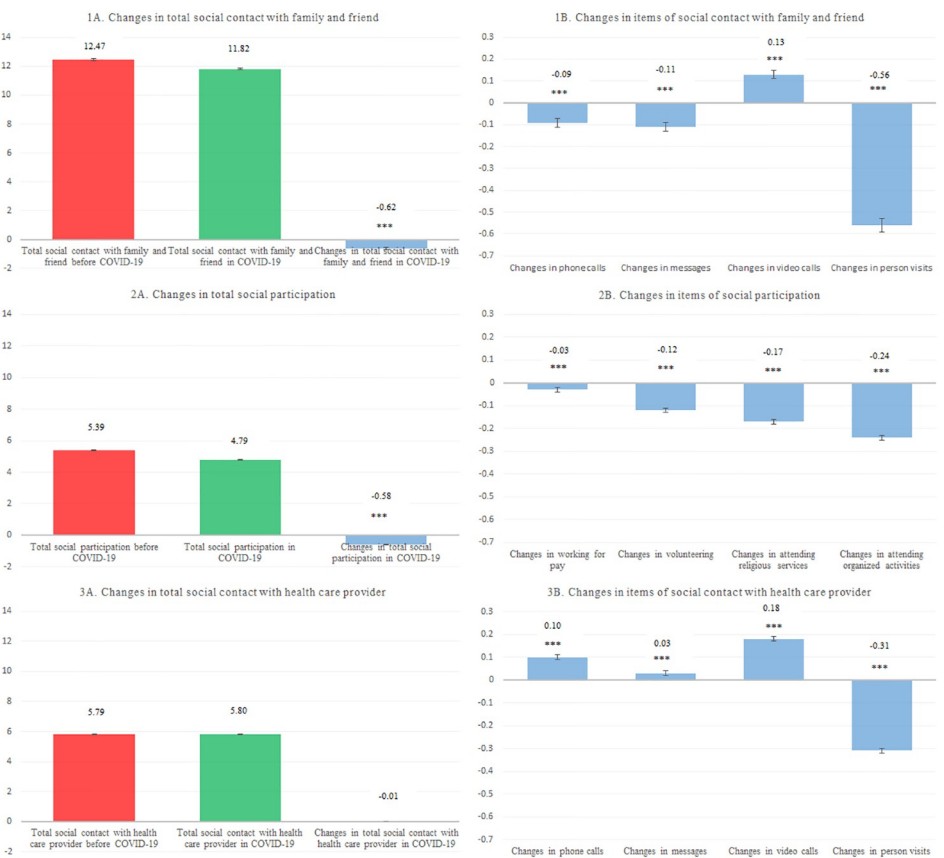

**Fig 1. Mean changes in self-reported social contact and social participation among older adults in the US during the Coronavirus (COVID-19) pandemic.** Notes: Analytic sample includes NHATS COVID-19 Questionnaire respondents who were self-respondents free of COVID-19 symptoms and diagnosis. The frequency of participants' social contact with family, friends, and healthcare providers, as well as social participation prior to and during the COVID-19 pandemic were compared using the paired t-test for summed scores. $p < 0.001$, Very significant, ***; $0.001 < = p < 0.01$, Very significant, **; $0.01 < = p < 0.05$, significant, *; $p \geq 0.05$. 1A and 1B Total social contact with family and friends refers to making phone calls; using emails, texts, or social media messages; making video calls, and in person visits with family and friends. 2A and 2B Total social contact with healthcare providers refers to making phone calls; using emails, texts, or social media messages; making video calls, and in person visits with healthcare provider. 3A and 3B Total social participation refers to working for pay, volunteering, attending religious services, and attending clubs, classes, or other organized activities. Items were assessed by a four-item 2-point scale.

To explore sociodemographic and health-related factors on changes in social contact and social participation during the COVID-19 pandemic, we presented multivariate linear regressions exploring the associations between changes in total social contact and social participation and various aforementioned factors among the US older adults in Table 2. Older age was associated with lower levels of total social contact with family and friends during the COVID-19 pandemic. When compared to females, older males had a significantly lower level of total social contact with family and friends (estimate: -0.66; $p < 0.0001$), but a higher level of total social contact with healthcare providers (estimate: 0.16; $p = 0.004$), during the COVID-19 pandemic. Non-Hispanic Black Americans exhibited higher levels of social contact with healthcare providers (estimate: 0.20; $p = 0.017$) and total social participation (estimate: 0.17; $p = 0.007$) during the COVID-19 pandemic, compared to non-Hispanic white Americans. Consistently, a lack of higher education was significantly associated with a lower level of total social contact with family and friends, total social contact with healthcare providers, and social participation,

**Table 2. Associations between sociodemographic and health factors with changes in total social contact and social participation among older adults in the US during the Coronavirus (COVID-19) pandemic.**

| | Total social contact with family and friends during COVID-19[1] | | Total social contact with family and friends before COVID-19[1] | | Changes in total social contact with family and friends during COVID-19[1] | |
|---|---|---|---|---|---|---|
| | Estimate (SE) | p | Estimate (SE) | p | Estimate (SE) | p |
| **Age group** | | | | | | |
| 75–79 years old | -0.32 (0.19) | 0.102 | -0.28 (0.19) | 0.151 | -0.19 (0.12) | 0.085 |
| 80–84 years old | -0.52 (0.21) | **0.014** | -0.56 (0.20) | **0.008** | -0.06 (0.12) | 0.596 |
| 85 years and older | -0.77 (0.23) | **0.001** | -0.95 (0.22) | **<0.0001** | 0.04 (0.13) | 0.753 |
| 70–74 years old | 0.00 | . | 0.00 | . | 0.00 | . |
| **Sex** | | | | | | |
| Male | -0.66 (0.14) | **<0.0001** | -0.86 (0.14) | **<0.0001** | 0.15 (0.08) | 0.071 |
| Female | 0.00 | . | 0.00 | . | 0.00 | . |
| **Race/ethnicity** | | | | | | |
| Black, non-Hispanic | 0.03 (0.20) | 0.890 | 0.08 (0.21) | 0.699 | 0.12 (0.12) | 0.320 |
| Other, non-Hispanic | -0.42 (0.54) | 0.368 | 0.06 (0.49) | 0.900 | -0.76 (0.32) | **0.004** |
| Hispanic | -0.20 (0.51) | 0.646 | -0.51 (0.48) | 0.233 | 0.27 (0.30) | 0.269 |
| White, non-Hispanic | 0.00 | . | 0.00 | . | 0.00 | . |
| **Education** | | | | | | |
| Less than high school | -1.33 (0.28) | **<0.0001** | -1.29 (0.27) | **<0.0001** | -0.08 (0.15) | 0.609 |
| High school or equivalent | -1.12 (0.18) | **<0.0001** | -0.89 (0.18) | **<0.0001** | -0.11 (0.10) | 0.277 |
| Beyond high school but less than college | -0.43 (0.17) | **0.011** | -0.46 (0.16) | **0.006** | -0.02 (0.09) | 0.804 |
| Bachelor and beyond | 0.00 | | 0.00 | . | 0.00 | . |
| **Marital status** | | | | | | |
| Separated/Divorced | -0.05 (0.20) | 0.811 | -0.21 (0.20) | 0.315 | 0.03 (0.13) | 0.810 |
| Widowed | 0.27 (0.17) | 0.123 | 0.13 (0.17) | 0.431 | 0.10 (0.10) | 0.289 |
| Never married | -1.28 (0.44) | **0.002** | -1.64 (0.43) | **<0.0001** | 0.38 (0.18) | 0.088 |
| Married/Living with a partner | 0.00 | . | 0.00 | . | 0.00 | . |
| **Earlier cognitive status** | | | | | | |
| Possible dementia | -0.47 (0.51) | 0.294 | -0.78 (0.52) | 0.083 | 0.44 (0.25) | 0.094 |
| Probable dementia | -0.23 (0.39) | 0.500 | -0.53 (0.39) | 0.130 | 0.27 (0.18) | 0.175 |
| No dementia | 0.00 | . | 0.00 | . | 0.00 | . |
| **SPPB score (physical performance)** | | | | | | |
| Low | -1.37 (0.21) | **<0.0001** | -1.42 (0.21) | **<0.0001** | 0.16 (0.12) | 0.168 |
| Medium | -0.53 (0.16) | **0.001** | -0.48 (0.16) | **0.003** | -0.04 (0.09) | 0.623 |
| High | 0.00 | . | 0.00 | . | 0.00 | . |
| **Number of chronic conditions[4]** | 0.06 (0.05) | 0.228 | 0.09 (0.05) | 0.099 | -0.02 (0.03) | 0.472 |
| | Total social contact with healthcare providers during COVID-19[2] | | Total social contact with healthcare providers before COVID-19[2] | | Changes in total social contact with healthcare providers during COVID-19[2] | |
| | Estimate (SE) | p | Estimate (SE) | p | Estimate (SE) | p |
| **Age group** | | | | | | |
| 75–79 years old | 0.01 (0.07) | 0.900 | -0.01 (0.06) | 0.806 | 0.03 (0.06) | 0.590 |
| 80–84 years old | -0.12 (0.08) | 0.107 | -0.10 (0.06) | 0.107 | -0.02 (0.07) | 0.816 |
| 85 years and older | -0.19 (0.08) | **0.026** | -0.12 (0.07) | 0.068 | -0.04 (0.07) | 0.573 |
| 70–74 years old | 0.00 | . | 0.00 | . | 0.00 | . |
| **Sex** | | | | | | |
| Male | 0.16 (0.05) | **0.004** | 0.06 (0.04) | 0.164 | 0.09 (0.04) | 0.054 |
| Female | 0.00 | . | 0.00 | . | 0.00 | . |

*(Continued)*

**Table 2.** (Continued)

| | Estimate (SE) | p | Estimate (SE) | p | Estimate (SE) | p |
|---|---|---|---|---|---|---|
| **Race/ethnicity** | | | | | | |
| Black, non-Hispanic | 0.20 (0.08) | **0.017** | 0.13 (0.07) | **0.048** | 0.07 (0.07) | 0.299 |
| Other, non-Hispanic | 0.16 (0.17) | 0.368 | 0.14 (0.15) | 0.330 | -0.07 (0.14) | 0.661 |
| Hispanic | 0.15 (0.14) | 0.303 | 0.16 (0.12) | 0.173 | -0.07 (0.12) | 0.564 |
| White, non-Hispanic | 0.00 | . | 0.00 | . | 0.00 | . |
| **Education** | | | | | | |
| Less than high school | -0.41 (0.09) | **<0.0001** | -0.37 (0.07) | **<0.0001** | -0.08 (0.07) | 0.364 |
| High school or equivalent | -0.34 (0.06) | **<0.0001** | -0.27 (0.05) | **<0.0001** | -0.07 (0.06) | 0.195 |
| Beyond high school but less than college | -0.15 (0.06) | **0.015** | -0.15 (0.05) | **0.003** | 0.01 (0.05) | 0.823 |
| Bachelor and beyond | 0.00 | . | 0.00 | . | 0.00 | . |
| **Marital status** | | | | | | |
| Separated/Divorced | -0.21 (0.08) | **0.010** | -0.01 (0.06) | 0.894 | -0.19 (0.07) | **0.005** |
| Widowed | -0.16 (0.06) | **0.011** | -0.07 (0.05) | 0.183 | -0.08 (0.05) | 0.152 |
| Never married | -0.37 (0.15) | **0.015** | -0.29 (0.11) | **0.017** | -0.02 (0.11) | 0.890 |
| Married/Living with a partner | 0.00 | . | 0.00 | . | 0.00 | . |
| **Earlier cognitive status** | | | | | | |
| Possible dementia | -0.12 (0.13) | 0.441 | 0.00 (0.12) | 0.997 | -0.03 (0.12) | 0.835 |
| Probable dementia | 0.08 (0.13) | 0.542 | -0.03 (0.09) | 0.756 | 0.07 (0.11) | 0.494 |
| No dementia | 0.00 | . | 0.00 | . | 0.00 | |
| **SPPB score (physical performance)** | | | | | | |
| Low | -0.05 (0.07) | 0.505 | -0.07 (0.06) | 0.233 | 0.05 (0.06) | 0.438 |
| Medium | -0.04 (0.06) | 0.541 | -0.02 (0.05) | 0.668 | 0.01 (0.05) | 0.836 |
| High | 0.00 | . | 0.00 | . | 0.00 | . |
| **Number of chronic conditions[4]** | 0.12 (0.02) | **<0.0001** | 0.05 (0.02) | **0.003** | 0.06 (0.22) | **<0.0001** |

| | Total social participation during COVID-19[3] | | Total social participation before COVID-19[3] | | Changes in total social participation during COVID-19[3] | |
|---|---|---|---|---|---|---|
| | **Estimate (SE)** | **p** | **Estimate (SE)** | **p** | **Estimate (SE)** | **p** |
| **Age group** | | | | | | |
| 75–79 years old | 0.05 (0.06) | 0.355 | 0.08 (0.07) | 0.228 | -0.03 (0.06) | 0.569 |
| 80–84 years old | -0.02 (0.06) | 0.788 | 0.06 (0.07) | 0.382 | -0.09 (0.07) | 0.189 |
| 85 years and older | -0.09 (0.06) | 0.164 | 0.01 (0.08) | 0.899 | -0.10 (0.07) | 0.154 |
| 70–74 years old | 0.00 | . | 0.00 | . | 0.00 | . |
| **Sex** | | | | | | |
| Male | 0.01 (0.04) | 0.790 | -0.33 (0.05) | **<0.0001** | 0.33 (0.04) | **<0.0001** |
| Female | 0.00 | . | 0.00 | . | 0.00 | . |
| **Race/ethnicity** | | | | | | |
| Black, non-Hispanic | 0.17 (0.06) | **0.007** | 0.08 (0.07) | 0.265 | 0.07 (0.06) | 0.243 |
| Other, non-Hispanic | -0.21 (0.15) | 0.141 | -0.29 (0.19) | 0.088 | 0.16 (0.19) | 0.291 |
| Hispanic | 0.04 (0.11) | 0.760 | 0.08 (0.15) | 0.568 | -0.05 (0.13) | 0.698 |
| White, non-Hispanic | 0.00 | . | 0.00 | . | 0.00 | . |
| **Education** | | | | | | |
| Less than high school | -0.31 (0.07) | **<0.0001** | -0.87 (0.08) | **<0.0001** | 0.55 (0.07) | **<0.0001** |
| High school or equivalent | -0.17 (0.05) | **0.001** | -0.55 (0.06) | **<0.0001** | 0.36 (0.06) | **<0.0001** |
| Beyond high school but less than college | -0.10 (0.06) | **0.046** | -0.31 (0.06) | **<0.0001** | 0.20 (0.06) | **0.000** |
| Bachelor and beyond | 0.00 | . | 0.00 | . | 0.00 | . |
| **Marital status** | | | | | | |
| Separated/Divorced | -0.18 (0.06) | **0.005** | -0.24 (0.08) | **0.001** | 0.13 (0.06) | 0.064 |
| Widowed | -0.02 (0.05) | 0.659 | 0.04 (0.06) | 0.460 | -0.06 (0.06) | 0.303 |

*(Continued)*

**Table 2.** (Continued)

| | | | | | | |
|---|---|---|---|---|---|---|
| Never married | 0.04 (0.14) | 0.724 | -0.15 (0.13) | 0.287 | 0.19 (0.11) | 0.137 |
| Married/Living with a partner | 0.00 | . | 0.00 | . | 0.00 | . |
| **Earlier cognitive status** | | | | | | |
| Possible dementia | -0.07 (0.11) | 0.599 | -0.23 (0.13) | 0.130 | 0.15 (0.10) | 0.266 |
| Probable dementia | -0.19 (0.08) | 0.058 | -0.26 (0.11) | **0.026** | 0.08 (0.10) | 0.476 |
| No dementia | 0.00 | . | 0.00 | . | 0.00 | . |
| **SPPB score (physical performance)** | | | | | | |
| Low | -0.22 (0.06) | **0.001** | -0.44 (0.07) | **<0.0001** | 0.23 (0.06) | **0.001** |
| Medium | -0.11 (0.05) | **0.019** | -0.14 (0.06) | **0.014** | 0.02 (0.05) | 0.692 |
| High | 0.00 | . | 0.00 | . | 0.00 | . |
| **Number of chronic conditions[4]** | -0.05 (0.02) | **0.003** | -0.07 (0.02) | **<0.0001** | 0.02 (0.02) | 0.279 |

**Notes**: Analytic sample includes NHATS COVID-19 Questionnaire respondents who were self-respondents free of COVID-19 symptoms and diagnosis. Multivariate linear regressions with Huber–White robust standard errors were used to investigate associations between socio-demographic, physical performances, physical performances, total social contact and total social participation among older adults in the US prior to and during the Coronavirus (COVID-19) pandemic and their changes. Bold typeface denotes statistical significance at $p < 0.05$.

[1] Total social contact with family and friends refers to making phone calls; using emails, texts, or social media messages; making video calls, and in person visits with family and friends.

[2] Total social contact with healthcare providers refers to making phone calls; using emails, texts, or social media messages; making video calls, and in person visits with healthcare providers.

[3] Total social participation refers to working for pay, volunteering, attending religious services, and attending clubs, classes, or other organized activities. Items were assessed by a four-item 2-point scale.

[4] The number of chronic conditions for each participant was calculated to summarize the burden of the following nine chronic conditions among the participants: heart attack, heart disease, high blood pressure, arthritis, osteoporosis, diabetes, lung disease, stroke, and cancer. Thus, the number of chronic conditions ranged from 0 to 9.

both prior to and during the COVID-19 pandemic. Compared to individuals who were married a and living with a spouse or living with partner, being separated/divorced, widowed, or never married was consistently associated with both a lower level of social contact with healthcare providers during the COVID-19 pandemic. Earlier cognitive performance was neither significantly associated with total social contact, nor social participation during the COVID-19 pandemic. However, lower physical performance was significantly correlated with a lower level of total social contact with family and friends, and total social contact with healthcare providers. During the COVID-19 pandemic, older adults with a larger number of chronic conditions exhibited a greater level of social contact with healthcare providers (estimate: 0.12; $p < 0.0001$) but a lower level of social participation (estimate: -0.05; p = 0.003).

We also explored associations between changes in social contact and social participation and the sociodemographic and health factors. Age was not significantly associated with changes in social participation or social contact during the COVID-19 pandemic. Interestingly, compared to females, males exhibited lesser social participation prior to the COVID-19 pandemic yet experienced a larger increase in the frequency of social participation (change: 0.33; p<0.0001) during the COVID-19 pandemic. Compared to non-Hispanic Whites, non-Hispanic other minorities reported a greater decrease in total social contact with family and friends during COVID-19 (change: -0.76; p<0.004). Notably, lower educational attainment was consistently associated with lower levels of social contact and social participation during the COVID-19 pandemic, but only higher increases in social participation during the pandemic. Compared to individuals who were married and living with a spouse or living with a partner, being separated/divorced was associated with a larger decrease in social contact with

healthcare providers during the COVID-19 pandemic (change: -0.19; p = 0.005). Prior cognitive status was not significantly associated with changes in total social contact and social participation. Older adults with low physical performance exhibited a significant increase in social participation during the COVID-19 pandemic (change: 0.23; p = 0.0001). Meanwhile, older adults with a larger number of chronic conditions exhibited significantly more social contact with healthcare providers during the COVID-19 pandemic (changes: 0.06; p = 0.0001).

To further assess how sociodemographic and health factors are associated with the observed change patterns in individual items of social contact and social participation, we used multivariate linear regressions and presented the results in Table 3. Participants in older age groups exhibited significantly smaller increases in the frequencies of video calls and smaller decreases in in-person visits. Meanwhile, the oldest-old participants (aged 85 years old and beyond) reported smaller decreases in working for pay, but larger decreases in volunteering and attending organized activities during the COVID-19 pandemic. Consistent with our observation that males exhibited significantly lower levels of social contact with friends and family, as well as with healthcare providers, during the COVID-19 pandemic, this effect was largely due to their smaller increases in video calls and a smaller decrease in in-person visits. Males also demonstrated significant increases in all four measures of social participation. Non-Hispanic other minorities showed a smaller increase in video calls with family and friends (changes: -0.23; p = 0.049), while non-Hispanic blacks showed a bigger increase in attending organized activities (changes: 0.07; p = 0.024) during the COVID-19 pandemic. Critically, older adults with lower educational attainment exhibited smaller increases in video calls and smaller decreases in in-person visits with friends and family or with healthcare providers when compared with participants with bachelor's degrees or higher. These individuals also demonstrated smaller decreases in changes in volunteering, attending religious services, and attending organized activities during the COVID-19 pandemic. Surprisingly, older adults with probable dementia exhibited a smaller decrease in the frequency of attending organized activities compared to older adults with no cognitive impairment. Older adults with the lowest level of prior physical performance exhibited smaller decreases in in-person visits with friends and family, volunteering, and organized activities during the COVID-19 pandemic. Older adults with a larger number of chronic conditions had smaller decreases in phone calls and in-person visits with healthcare providers, but larger increases in video calls.

## Discussion

During the COVID-19 pandemic, older adults in the US exhibited significant decreases in social contact with family and friends and social participation, but not in social contact with healthcare providers. Notably, older adults exhibited a decrease in the use of non-visual electronic forms of messaging (such as emails, texts, phone calls) and in-person visits, accompanied by a considerable increase in the use of video calls, in contact with family and friends, and with healthcare providers. Older age groups, lower educational attainment, lack of a spouse/partner, and poorer physical performance, were associated with lower levels of social contact with family and friends, and with healthcare provider. Especially, Participants who were older, males, and lower educational attainment, exhibited significantly smaller increases in the frequencies of video calls and smaller decreases in in-person visits. Meanwhile, lower educational attainment, poorer physical performance, and a larger number of chronic conditions were associated with less social participation.

With digital forms of communication and pandemic-induced social isolation transforming the social landscape, it is necessary to understand how social contact and social participation have changed across populations and to describe how older adults have responded to social

**Table 3. Associations between socio-demographic, health factors and changes in individual measures of social contact and social participation among older adults in the US during the Coronavirus (COVID-19) pandemic.**

| | Changes in phone calls with family and friend in COVID-19 | | Changes in message with family and friend in COVID-19 | | Changes in video calls with family and friend in COVID-19 | | Changes in person visits with family and friend in COVID-19 | | Changes in phone calls with healthcare provider in COVID-19 | | Changes in message with healthcare provider in COVID-19 | |
|---|---|---|---|---|---|---|---|---|---|---|---|---|
| | E* | p | E* | p | E* | p | E* | p | E* | p | E* | p |
| **Age group** | | | | | | | | | | | | |
| 75–79 years old | -0.01 (0.04) | 0.890 | -0.02 (0.05) | 0.696 | -0.12 (0.05) | **0.012** | -0.01 (0.07) | 0.895 | 0.00 (0.03) | 0.940 | 0.03 (0.02) | 0.075 |
| 80–84 years old | -0.05 (0.05) | 0.268 | -0.02 (0.05) | 0.663 | -0.12 (0.05) | **0.026** | 0.15 (0.07) | **0.032** | 0.01 (0.03) | 0.836 | 0.01 (0.02) | 0.619 |
| 85 years and older | -0.04 (0.05) | 0.438 | 0.01 (0.05) | 0.907 | -0.14 (0.06) | **0.012** | 0.21 (0.08) | **0.006** | -0.01 (0.03) | 0.695 | 0.00 (0.02) | 0.991 |
| 70–74 years old | 0.00 | . | 0.00 | . | 0.00 | . | 0.00 | . | 0.00 | . | 0.00 | . |
| **Sex** | | | | | | | | | | | | |
| Male | -0.01 (0.03) | 0.702 | 0.03 (0.04) | 0.375 | -0.09 (0.04) | **0.011** | 0.20 (0.05) | **<0.0001** | 0.03 (0.02) | 0.191 | 0.02 (0.02) | 0.240 |
| Female | 0.00 | . | 0.00 | . | 0.00 | . | 0.00 | . | 0.00 | . | 0.00 | . |
| **Race/ethnicity** | | | | | | | | | | | | |
| Black, non-Hispanic | 0.08 (0.05) | 0.101 | 0.02 (0.06) | 0.694 | -0.06 (0.05) | 0.255 | 0.11 (0.06) | 0.129 | 0.01 (0.03) | 0.754 | 0.04 (0.03) | 0.111 |
| Other, non-Hispanic | -0.06 (0.15) | 0.583 | -0.13 (0.12) | 0.232 | -0.23 (0.09) | **0.049** | -0.13 (0.14) | 0.400 | -0.03 (0.07) | 0.661 | 0.03 (0.04) | 0.574 |
| Hispanic | 0.16 (0.13) | 0.075 | 0.04 (0.13) | 0.700 | 0.14 (0.15) | 0.197 | -0.09 (0.18) | 0.510 | -0.06 (0.06) | 0.309 | 0.04 (0.05) | 0.290 |
| White, non-Hispanic | 0.00 | . | 0.00 | . | 0.00 | . | 0.00 | . | 0.00 | . | 0.00 | . |
| **Education** | | | | | | | | | | | | |
| Less than high school | -0.03 (0.06) | 0.640 | -0.06 (0.07) | 0.314 | -0.26 (0.06) | **<0.0001** | 0.32 (0.09) | **0.0003** | -0.02 (0.04) | 0.631 | -0.03 (0.02) | 0.279 |
| High school or equivalent | -0.02 (0.04) | 0.605 | -0.04 (0.04) | 0.296 | -0.28 (0.04) | **<0.0001** | 0.24 (0.06) | **<0.0001** | 0.01 (0.03) | 0.800 | -0.02 (0.02) | 0.243 |
| Beyond high school but less than college | -0.02 (0.04) | 0.608 | 0.01 (0.04) | 0.893 | -0.20 (0.04) | **<0.0001** | 0.20 (0.06) | **0.001** | 0.01 (0.03) | 0.702 | -0.02 (0.02) | 0.349 |
| Bachelor and beyond | 0.00 | . | 0.00 | . | 0.00 | . | 0.00 | . | 0.00 | . | 0.00 | . |
| **Marital status** | | | | | | | | | | | | |
| Separated/Divorced | 0.01 (0.06) | 0.891 | -0.01 (0.06) | 0.875 | 0.01 (0.06) | 0.845 | 0.04 (0.08) | 0.540 | -0.04 (0.03) | 0.189 | -0.03 (0.02) | 0.132 |
| Widowed | 0.08 (0.04) | **0.036** | 0.05 (0.04) | 0.176 | -0.03 (0.04) | 0.471 | 0.01 (0.06) | 0.824 | -0.02 (0.03) | 0.537 | -0.01 (0.02) | 0.525 |
| Never married | 0.07 (0.08) | 0.428 | 0.06 (0.07) | 0.532 | -0.10 (0.05) | 0.309 | 0.37 (0.10) | **0.008** | 0.00 (0.06) | 0.979 | 0.07 (0.05) | 0.082 |
| Married/Living with a partner | 0.00 | . | 0.00 | . | 0.00 | . | 0.00 | . | 0.00 | . | 0.00 | . |
| **Earlier cognitive status** | | | | | | | | | | | | |
| Possible dementia | 0.06 (0.10) | 0.571 | 0.06 (0.06) | 0.544 | 0.06 (0.12) | 0.585 | 0.13 (0.13) | 0.388 | -0.08 (0.06) | 0.259 | -0.01 (0.03) | 0.874 |
| Probable dementia | 0.10 (0.08) | 0.194 | 0.13 (0.10) | 0.119 | 0.09 (0.12) | 0.326 | -0.14 (0.11) | 0.237 | -0.03 (0.05) | 0.598 | 0.01 (0.03) | 0.800 |
| No dementia | 0.00 | . | 0.00 | . | 0.00 | . | 0.00 | . | 0.00 | . | 0.00 | . |
| **SPPB score (physical performance)** | | | | | | | | | | | | |
| Low | -0.01 (0.05) | 0.833 | 0.04 (0.05) | 0.430 | -0.05 (0.05) | 0.296 | 0.14 (0.07) | **0.049** | 0.02 (0.03) | 0.448 | 0.01 (0.02) | 0.575 |
| Medium | 0.00 (0.04) | 0.945 | -0.03 (0.03) | 0.460 | -0.05 (0.04) | 0.182 | 0.00 (0.06) | 0.959 | 0.02 (0.03) | 0.530 | -0.01 (0.02) | 0.658 |
| High | 0.00 | . | 0.00 | . | 0.00 | . | 0.00 | . | 0.00 | . | 0.00 | . |
| **Number of chronic diseases[1]** | -0.01 (0.01) | 0.514 | 0.01 (0.01) | 0.537 | 0.00 (0.01) | 0.728 | 0.00 (0.02) | 0.785 | 0.02 (0.01) | **0.042** | 0.00 (0.00) | 0.594 |

*(Continued)*

**Table 3.** (Continued)

| | Changes in video calls with healthcare provider in COVID-19 | | Changes in person visits with healthcare provider in COVID-19 | | Changes in working for pay in COVID-19 | | Changes in volunteering in COVID-19 | | Changes in attending religious services in COVID-19 | | Changes in attending organized activities in COVID-19 | |
|---|---|---|---|---|---|---|---|---|---|---|---|---|
| | E* | p | E* | p | E* | p | E* | p | E* | p | E* | p |
| **Age group** | | | | | | | | | | | | |
| 75–79 years old | 0.00 (0.03) | 0.853 | 0.02 (0.03) | 0.610 | 0.01 (0.02) | 0.472 | -0.03 (0.02) | 0.123 | 0.01 (0.03) | 0.650 | -0.03 (0.03) | 0.216 |
| 80–84 years old | -0.05 (0.03) | 0.115 | 0.05 (0.03) | 0.150 | 0.01 (0.02) | 0.516 | -0.04 (0.02) | 0.081 | -0.02 (0.03) | 0.460 | -0.05 (0.03) | 0.136 |
| 85 years and older | -0.07 (0.03) | **0.025** | 0.08 (0.03) | **0.029** | 0.04 (0.02) | **0.013** | -0.01 (0.02) | 0.661 | -0.08 (0.03) | **0.007** | -0.07 (0.03) | **0.023** |
| 70–74 years old | 0.00 | . | 0.00 | . | 0.00 | . | 0.00 | . | 0.00 | . | 0.00 | . |
| **Sex** | | | | | | | | | | | | |
| Male | -0.06 (0.02) | **0.006** | 0.08 (0.02) | **0.0002** | 0.02 (0.01) | **0.030** | 0.09 (0.02) | **<0.0001** | 0.06 (0.02) | **0.003** | 0.15 (0.02) | **<0.0001** |
| Female | 0.00 | . | 0.00 | . | 0.00 | . | 0.00 | . | 0.00 | . | 0.00 | . |
| **Race/ethnicity** | | | | | | | | | | | | |
| Black, non-Hispanic | 0.01 (0.03) | 0.683 | 0.06 (0.03) | 0.071 | 0.00 (0.01) | 0.853 | 0.02 (0.02) | 0.305 | -0.02 (0.03) | 0.432 | 0.07 (0.03) | **0.024** |
| Other, non-Hispanic | -0.07 (0.07) | 0.306 | -0.01 (0.09) | 0.876 | 0.02 (0.04) | 0.452 | 0.05 (0.06) | 0.385 | 0.04 (0.07) | 0.511 | 0.02 (0.08) | 0.811 |
| Hispanic | -0.01 (0.05) | 0.834 | -0.05 (0.07) | 0.468 | -0.02 (0.03) | 0.538 | -0.01 (0.04) | 0.770 | 0.04 (0.05) | 0.457 | -0.04 (0.05) | 0.549 |
| White, non-Hispanic | 0.00 | . | 0.00 | . | 0.00 | . | 0.00 | . | 0.00 | . | 0.00 | . |
| **Education** | | | | | | | | | | | | |
| Less than high school | -0.13 (0.03) | **0.001** | 0.12 (0.04) | **0.004** | 0.02 (0.01) | 0.262 | 0.14 (0.03) | **<0.0001** | 0.15 (0.03) | **<0.0001** | 0.23 (0.03) | **<0.0001** |
| High school or equivalent | -0.11 (0.02) | **<0.0001** | 0.07 (0.03) | **0.016** | 0.01 (0.01) | 0.228 | 0.13 (0.02) | **<0.0001** | 0.07 (0.02) | **0.004** | 0.13 (0.03) | **<0.0001** |
| Beyond high school but less than college | -0.04 (0.02) | 0.123 | 0.07 (0.03) | **0.004** | 0.02 (0.01) | 0.164 | 0.06 (0.02) | **0.001** | 0.05 (0.02) | **0.017** | 0.08 (0.03) | **0.002** |
| Bachelor and beyond | 0.00 | . | 0.00 | . | 0.00 | . | 0.00 | . | 0.00 | . | 0.00 | . |
| **Marital status** | | | | | | | | | | | | |
| Separated/Divorced | -0.04 (0.03) | 0.146 | -0.05 (0.03) | 0.123 | -0.02 (0.02) | 0.268 | 0.05 (0.02) | **0.041** | 0.03 (0.03) | 0.271 | 0.05 (0.03) | 0.135 |
| Widowed | -0.05 (0.02) | **0.049** | -0.02 (0.03) | 0.563 | -0.02 (0.01) | 0.055 | -0.03 (0.02) | 0.105 | 0.00 (0.02) | 0.839 | 0.00 (0.03) | 0.878 |
| Never married | 0.06 (0.06) | 0.298 | -0.14 (0.07) | **0.022** | -0.02 (0.02) | 0.542 | 0.08 (0.03) | 0.097 | 0.11 (0.04) | **0.045** | -0.01 (0.06) | 0.883 |
| Married/Living with a partner | 0.00 | . | 0.00 | . | 0.00 | . | 0.00 | . | 0.00 | . | 0.00 | . |
| **Earlier cognitive status** | | | | | | | | | | | | |
| Possible dementia | -0.02 (0.06) | 0.756 | 0.02 (0.07) | 0.815 | 0.02 (0.03) | 0.584 | 0.06 (0.05) | 0.224 | 0.06 (0.05) | 0.313 | 0.04 (0.05) | 0.529 |
| Probable dementia | 0.08 (0.05) | 0.109 | -0.01 (0.06) | 0.837 | -0.01 (0.02) | 0.696 | 0.00 (0.03) | 0.912 | -0.01 (0.05) | 0.824 | 0.11 (0.04) | **0.029** |
| No dementia | 0.00 | . | 0.00 | . | 0.00 | . | 0.00 | . | 0.00 | . | 0.00 | . |
| **SPPB score (physical performance)** | | | | | | | | | | | | |
| Low | -0.04 (0.03) | 0.143 | 0.05 (0.03) | 0.089 | 0.02 (0.01) | 0.152 | 0.06 (0.02) | **0.022** | 0.03 (0.03) | 0.354 | 0.13 (0.03) | **<0.0001** |
| Medium | -0.02 (0.02) | 0.343 | 0.01 (0.02) | 0.766 | 0.01 (0.01) | 0.401 | 0.00 (0.02) | 0.951 | -0.01 (0.02) | 0.656 | 0.03 (0.02) | 0.278 |
| High | 0.00 | . | 0.00 | . | 0.00 | . | 0.00 | . | 0.00 | . | 0.00 | . |
| **Number of chronic diseases**[1] | 0.02 (0.01) | **0.010** | 0.03 (0.01) | **0.001** | 0.00 (0.00) | 0.619 | 0.01 (0.01) | 0.348 | 0.00 (0.01) | 0.569 | 0.01 (0.01) | 0.328 |

**Notes**: Analytic sample includes NHATS COVID-19 Questionnaire respondents who were self-respondents free of COVID-19 symptoms and diagnosis. Multivariate linear regressions with Huber–White robust standard errors were used to investigate associations between socio-demographic, physical performances, and individual items of social contact and social participation among older adults in the US prior to and during the Coronavirus (COVID-19) pandemic and their changes. Bold typeface denotes statistical significance at $p < 0.05$.

[1] The number of chronic conditions for each participant was calculated to summarize the burden of the following nine chronic conditions among the participants: heart attack, heart disease, high blood pressure, arthritis, osteoporosis, diabetes, lung disease, stroke, and cancer. Thus, the number of chronic conditions ranged from 0 to 9.

crises. Consistent with prior studies [29, 30], our study reported that total social contact with friends and family and total social participation among older adults, decreased significantly during the COVID-19 pandemic (Fig 1). Observed changes in social contact might be attributed to large decreases in in-person visits with friends and family, along with larger increase in video calls and a considerable increase in the use of non-visual electronic forms of messaging (email, text, phone calls). The decrease in in-person visits is largely due to gathering restrictions and social distancing measures imposed during the COVID-19 pandemic, which encourages a shifting trend from larger group interactions with more distant friends and family to more limited interactions with closer friends and family. Our findings suggest that the lack of digital literacy amongst older adults especially hinders the maintenance of distant digital ties with friends and family after the loss of in-person contact.

A study conducted by Freedman and colleagues [31] reported changes in weekly contact with family and friends and stressed the importance of using communication technologies during the pandemic times. Although using the same NHATS data, our research has a different set of inclusion/exclusion criteria, and different variables to stratify the sample. Our results extend previous work to show that COVID-19 induced changes in social contact with healthcare providers and social participation of older populations, and also describe how sociodemographic and health factors influenced those changes.

We found that the frequency of in-person visits decreased, while the frequency of all digital forms of contact with healthcare providers increased among older adults during the COVID-19 pandemic. This supports the findings of previous literature investigating hospital admissions in the US during the COVID-19 pandemic, which reported dramatic decreases in non-COVID-19-related in-person hospital visits [17, 32]. Many adults, particularly older adults, ceased making regular in-person visits to healthcare providers for chronic conditions, annual checkups, and even acute medical illness in fear of nosocomial contracting COVID-19 or long hospital waiting periods [33]. Notably, results indicate that older adults with more chronic conditions experienced similar decreases in total social contact with healthcare providers compared to older adults with lesser chronic conditions, further highlighting the critical loss of social contact with healthcare providers among particularly vulnerable populations during the COVID-19 pandemic. Changes in clinical availability throughout the pandemic further exacerbated the loss of clinical contact induced by social distancing measures and fear of nosocomial disease, as many clinical services were halted or significantly reduced because of COVID-related shutdowns. As an alternative, many healthcare providers implemented remote healthcare visitations known as "telehealth", through which doctors could meet with patients from home, thereby maintaining safer contact with vulnerable older patients during the COVID-19 pandemic.

Our study found that the frequencies of all forms of social participation, including working for pay, volunteering, attending religious services, and attending organized activities during the COVID-19 pandemic, decreased. Our data did not separate remote and physical forms of social participation, so this observed decrease in social participation is especially notable given the advent of digital platforms, as an online space where exchanges of information and services to occur between individuals, groups, and community. Many social organizations, classes, and churches began to offer virtual platforms during the COVID-19 pandemic, increasing their accessibility and making it possible for people in one geographic location to attend social activities that used to be restricted due to location barriers. In response to large-scale shifts from in-person to digital social platforms, social organizations and healthcare providers have implemented new digital literacy courses [34, 35], older adults-targeted online social gatherings [36, 37], and digital exercise programs [38] to promote the use of technology for social interaction amongst older adults. Despite this growing

movement for digital literacy, the digital divide between older and younger populations exacerbates the vulnerable social state of older Americans in this new digital age and remains a major barrier to providing adequate social support for older adults during the COVID-19 pandemic [34, 39]. The pronounced decrease in social participation for older adults during the COVID-19 pandemic amidst convenient forms of digital communication suggests that, despite increased efforts to promote digital literacy, new digital platforms for social activities, such as remote church gatherings and remote classes, are being inadequately utilized by older populations. This, coupled with findings reporting that digital literacy improves quality of life and reduces feelings of social isolation [37, 40] emphasizes the critical importance of promoting digital literacy and providing accessible digital platforms for older adults to maintain social contact and social participation, especially in a major crisis. To help mitigate the digital divide, healthcare providers and social organizations could invest in digital programs made specifically for older people such as: digital skills training, online exercise programs, digital social gatherings, promote access to affordable forms of technology, and advertise digital literacy courses to older populations.

Our study reported that sex and education level were associated with social contact and social participation during the COVID-19 pandemic. Consistent with previous literature reporting that older female adults tend to have more social ties and social interactions than older male adults [21], we found that males reported lower levels of social contact and social participation both prior to and during the COVID-19 pandemic. Compared to females, males had smaller decreases in social contact and social participation; they tended to make fewer video calls but more in-person visits to contact with family and friends. Another major determinant of social contact and social participation that our study identified was education. Attainment of a higher educational level was associated with higher levels of social contact with family, friends, and healthcare providers, along with more social participation prior to and during the COVID-19 pandemic. Older adults with higher educational degrees had larger increases in video calls and larger decreases in in-person visits with friends and family, as well as with healthcare providers, suggesting that older adults with higher educational degrees may be both less inclined to make in-person visits to adhere to social guidelines and more digitally literate [41, 42].

Our research also found that prior physical performance, rather than prior cognitive ability, is significantly positively associated with social contact with family and friends and social participation. These relationships highlight a potentially dangerous vicious cycle of physical health and social participation, wherein the loss of social participation because of the COVID-19 pandemic hinders the physical health of older adults, which in turn further decreases social participation [43].

As a result of pandemic-induced restrictions and shifting means of communication, promoting "active aging"—the ability of older adults to independently maintain healthy social, economic, cultural, spiritual, and civic affairs—has rapidly become a critical medical and societal challenge. Especially, the COVID-19 pandemic has challenged the most traditional and popular ways to use social contact and social participation to link older adults and social inclusion, to solve the problems of social exclusion and isolation of the older adults. Our research emphasizes the importance and possibility of using digital ways to keep older adults socially active in touch with family, friends, and healthcare provider, and engaged in social activities, without affecting social distancing. Previous research has summarized three levels of digital divides, including (1) material access, (2) skills and uses, and (3) outcomes of differentiated access and use [44]. Although this research focused on how old Americans used digital access to involve social contact and social participation, we should not ignore the basic inequities and divides in equipment, internet access, and skills.

There are several strengths of our research. Firstly, our study incorporated multidimensional measures of social contact and social participation, analyzing both aggregated and individual data. This allowed us to identify detailed trends regarding changes in social contact with family and friends, and healthcare providers and social participation and social platforms that were most affected during the COVID-19 pandemic, providing information that enables policymakers and social organizations to efficiently promote the social wellbeing of older adults through more cost-effective and targeted interventions. Furthermore, unlike studies utilizing a cross-sectional study design, the use of pre-COVID data allows us to contribute to some understanding of the causative relationships between physical and cognitive function and social patterns of older adults.

There are also several limitations in our study and opportunities for further work. Firstly, our study used the NHATS protocol—a combination of screening interviews, cognitive tests, and self-reported or proxy-reported physician diagnosis of dementia—to classify participants as no dementia, possible dementia, or probable dementia. Although the protocol correlates strongly with dementia, it cannot adequately substitute for clinical assessments. Meanwhile, self-reported data on social contact and social participation prior to and during the COVID-19 pandemic were collected through the NHATS COVID-19 questionnaire, which was administered during the COVID-19 pandemic (June 2020—November 2020) and may be subject to recall bias. Furthermore, the NHATS COVID-19 questionnaire was self-administered and subject to nonresponse bias, with response rates varying significantly by sociodemographic status. Thus, most of our participants were non-Hispanic White and relatively highly educated participants. Additionally, most participants reported non-Hispanic White race and were relatively well-educated, making them more likely to have access to technology within their home. Thus, the reported experience and challenges of our participants during the COVID-19 period may not represent the experience of older adults who lacked such facile access, which limits the generalizability of the findings.

## Conclusion

This study reports that the frequency of total social contact with family and friends, as well as in-person and virtual social participation, decreased significantly during the COVID-19 pandemic among older adults in the US. Moreover, age, sex, education, and physical impairment prior to the introduction of COVID-19 in the U.S. were significant risk factors for lowered social contact. Our research might therefore suggest the implementation of digital social gatherings could be useful for older adults. This study might also suggest the need for the development of accessible tele-healthcare programs, potentially designed with older vulnerable adults in mind, to facilitate healthcare access as we continue to respond to seasonal endemic diseases like COVID and influenza that are more severe in the elderly.

## Author Contributions

**Conceptualization:** Yun Zhang, Sean A. P. Clouston.

**Formal analysis:** Yun Zhang.

**Methodology:** Yun Zhang, Luke Hou, Wei Zhang, Andrew Schwartz.

**Software:** Luke Hou.

**Supervision:** Sean A. P. Clouston.

**Visualization:** Yun Zhang, Shanquan Chen, Andrew Schwartz.

**Writing – original draft:** Yun Zhang.

**Writing – review & editing:** Yun Zhang, Amber Luo, Shanquan Chen, Wei Zhang, Sean A. P. Clouston.

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
