## [Decision Letter · Decision Letter 0]

4 Nov 2022

PONE-D-22-10689Crisis Response During the COVID-19 Pandemic: Changes in Social Contact and Social Participation of Older Americans in the US.PLOS ONE

Dear Dr. Clouston,

Thank you for submitting your manuscript to PLOS ONE. Please accept my sincere apologies for the delay in hearing back.  After careful consideration, we feel that it has merit but does not fully meet PLOS ONE’s publication criteria as it currently stands. Therefore, we invite you to submit a revised version of the manuscript that addresses the points raised during the review process.

A reviewer highlighted several areas where the manuscript may benefit from revisions on your part -- particularly those regarding clarity and the last.  Please find the report attached.  I would also suggest you consider being clearer about who comprises the reference group as you discuss your regression results.

We look forward to receiving your revised manuscript.

Kind regards,

Michael E Martell

Academic Editor

PLOS ONE

Journal Requirements:

Reviewers' comments:

Reviewer's Responses to Questions

**Comments to the Author**

1. Is the manuscript technically sound, and do the data support the conclusions?

Reviewer #1: Yes

2. Has the statistical analysis been performed appropriately and rigorously? 

Reviewer #1: N/A

3. Have the authors made all data underlying the findings in their manuscript fully available?

Reviewer #1: Yes

4. Is the manuscript presented in an intelligible fashion and written in standard English?

Reviewer #1: Yes

5. Review Comments to the Author

Reviewer #1: The statistical analysis seems appropriate; however, I am not a statistician; but the use of t-test to compare categorical data seems appropriate. The descriptive data seems appropriate as well and captures the amount of participants included in this secondary data analysis.

6. PLOS authors have the option to publish the peer review history of their article (what does this mean?). If published, this will include your full peer review and any attached files.

Reviewer #1: No

---

## [Author Response · Author response to Decision Letter 0]

22 Dec 2022

PONE-D-22-10689

Crisis Response During the COVID-19 Pandemic: Changes in Social Contact and Social Participation of Older Americans in the US.

Response to Reviewers (Major Revision)

Dear Dr. Martell,

Thank you for the opportunity to further improve our paper to fully meet PLOS ONE’s publication criteria. We appreciate the reviewers’ time and careful attention, which have been very helpful in improving our manuscript. Below we address all the raised feedback point by point, upon which our manuscript has been polished with track changes. Furthermore, we have carefully edited the manuscript to catch any remaining issues and make sure that our manuscript meets PLOS ONE's style requirements. 

Reviewers' Comments to the Authors:

REVIEWER 1

Thank you for allowing me the opportunity to read this paper and provide comments.

COVID-19 will continue to be a topic we will grapple with for years to come. This is especially relevant as we concern ourselves with the health and well-being of older adults in America, which is the fastest growing segment of our population. COVID-19 has impacted this group the most and in ways that we would not have imagined pre-COVID-19.

Comment 1. Objective: Line 35-37 The authors indicated assessment of two aims, 1) changes in social contact and participation, and 2) roles of socio-demographics and health-related factors. It would be helpful if the authors provided their definition of 1) social contact and 2) social participation so that it is clear from the outset what the authors plan to measure. As it written, it seems as these two constructs could be the same. Secondly, when the author indicates the roles of socio-demographics and health-related factors in relations to what. The authors might revise so that the audience clearly understand the points of your study. As I read the document, sometimes this was clear, most times it was not.

Response: We added the definition of social contact and social participation into the abstract as suggested. We also polished the manuscript to clarify that our second aim was to explore the roles of sociodemographic, cognitive, and physical conditions in relation to social contact and participation.

Comment 2. Discussion: Line 52-53. The authors must link this promoting digital literacy and providing access to digital platforms better to social contact and social participation. As written, its digital literacy had not been discussed. 

Response: We have now removed a reference to digital literacy and comment, instead, on the general role of telehealth in this population, noting (page 22) that “This study might also suggest the need for the development of accessible forms of telehealth care to facilitate healthcare access in vulnerable populations of older adults.” 

Comment 3. Line 62: should be commas after, facilities and homes

Response: Corrected.

Comment 4. Line 63-65: Reviewer appreciate the definition of social contact and participation here; but recommend it in the abstract as well.

Response: Revised.

Comment 5. Line 74-75: Reviewer appreciate the first mention of digital literacy in relations to social contact; but recommend mentioning this term in the abstract.

Response: Revised.

Comment 6. Line 98-102: Author’s goals as written here seems to be what the authors might convey in their abstract for objective 2. It is clear and provides the reader with what is to be measured and description of change.

Response: We have revised the abstract accordingly.

Comment 7. Line 131: The term eventually seems an odd phrase. As I add the numbers 414 and 288 to 2486 the total is 3188, which would make this a final result. I would suggest revising.

Response: Revised.

Comment 8. Line 149-150: If I am reading this correctly, it appears that no demographic information was asked of participants in Round 10 but the authors went to Round 9 database and matched the demographics of participants to Round 10. It is unclear how this was done and might be worth a short explanation of this.

Response: Thank you for pointing this out. We have incorporated your suggestion in the methods section to clarify that “Using NHATS’ longitudinal tracking identifiers, we linked NHATS COVID-19 Questionnaire responses to data collected in Round 9 to measure pre-COVID physical and cognitive impairments and to identify sociodemographic information.

Comment 9. Line 164: Was there a specific test that NHATS used to determine dementia status, or did NHATS develop its own test.

Response: The NHATS has its own protocol for diagnosing dementia for research purposes but uses standard tests that are common across research studies and are included in clinical tools such as the TICS, MMSE, and MoCA exams. As now described in the “Cognitive performance” section, NHATS assessed cognitive performance using: “(1) a self-reported or proxy-reported physician diagnosis of Alzheimer’s Disease and dementias; (2) an informant-reported score indicating probable dementia on the AD8 Dementia Screening Interview; and (3) a series of cognitive tests for orientation, memory, and executive functioning”. We now also describe the NHATS dementia status criteria, on which we classified the cognitive status of participants into probable dementia, possible dementia, and no dementia. 

Comment 10. Line 204: The reviewer suggests that when writing the p value, it should be written consistently throughout the document. I some places the “P” is capitalized and in other places it is not. In most research manuscript the p value is not capitalized.

Response: Thank you for the kind reminder. We now consistently use uncapitalized “p” values.

Comment 11. Line 262: Authors use the term lesser social involvement is this the same as social contact?

Response: Thank for the question. We have polished the manuscript to clarify.

Comment 12. Line 266: Authors use the term lower education degrees is this the same as having a high school degree?

Response: We have shifted the manuscript to discuss the general trend, using the more standard term “lower educational attainment”. 

Comment 13. Line 379-382: While authors make a great recommendation of what healthcare providers and other social organizations should do, there are limitations that the older adults does not have access to internet, older adults are not digitally literate (unable to operate a computer), and older adults do not have computers in their homes.

Response: We agree and have incorporated your suggestion to discuss the importance of “social inclusion” in the discussion section on page 19. 

Comment 14. Line 424: The authors earlier used the term “no dementia” the authors here use the term “cognitively usual”. Reviewer recommends consistent use of terms.

Response: Thank you for the advice. We agree and have revised our manuscript to consistently use the term “no dementia”.

Comment 15. Line 428-433: Reviewer believes this makes my final point. The participants in this study were white, highly educated, and more likely have access to technology within their home. This leads to my conclusion, while this study is helpful to determine policy; but how does this help those who are poor, less educated, and limited to access to technology, increase their quality of social contact or social participation. It seems the group that was most prevalent in this study already had a significant advantage and policies related to making cities more “age friendly” will only add to this advantage for this group.

Response: We agree and now state that: “most participants reported non-Hispanic White race and were relatively well-educated, making them more likely to have access to technology within their home” and have added this as a limitation of our research. However, we also notice that some individuals in our sample reported other races including Hispanic or non-Hispanic Black and many lack a high school degree supporting the conclusion that the cohort is generalizable. We agree that an age-inclusive and age-friendly society could improve outcomes in general and have modified the discussion, in part to address this conclusion.

---

## [Editor Report · Decision Letter 1]

10 Jan 2023

PONE-D-22-10689R1Crisis Response During the COVID-19 Pandemic: Changes in Social Contact and Social Participation of Older Americans in the US.PLOS ONE

Dear Dr. Clouston,

Thank you for submitting your manuscript to PLOS ONE. After careful consideration, we feel that it has merit but does not fully meet PLOS ONE’s publication criteria as it currently stands. Therefore, we invite you to submit a revised version of the manuscript that addresses the points raised during the review process. The manuscript is improved as a result of your revisions in response to the reviewer.  However, there are areas where it appears the reviewer comments has not yet been adequately resolved and some revisions decreased clarity.  Please find my comments on these below and respond to each.

We look forward to receiving your revised manuscript.

Kind regards,

Michael E Martell

Academic Editor

PLOS ONE

Journal Requirements:

Additional Editor Comments:

1) The abstract (under objective) should define  participation.  Exactly, participation in what?  

2) Please clarify informed consent on page 8.  Does the manuscript now explain that the survey administrators did not collect consent?

3) In describing the regression approach, clarify if/that standard errors were adjusted to correct for heteroscedasticity.

4) Please respond more fully to reviewer comment 15, re: acknowledging the impact of the lack of representativeness in the sample impact on your ability to generalize results.  I believe the current claim oversells.  This discussion in the manuscript should include other demographic characteristics (such as race) in addition to education.  If the you wish to claim more generalizability, the manuscript needs to utilize and engage with weighting estimates at a minimum.

---

## [Author Response · Author response to Decision Letter 1]

27 Mar 2023

PONE-D-22-10689R1

EMID:64fa3d01b4287516

Crisis Response During the COVID-19 Pandemic: Changes in Social Contact and Social Participation of Older Americans in the US.

Response to Reviewers (Minor Revision)

Dear Dr. Martell,

We appreciate the chance to make further revisions to our paper for the PLOS ONE Journal and would like to express our gratitude for your thoughtful and meticulous feedback. Our team has taken great care to address each comment and we believe that the additional revisions have significantly enhanced the quality of our manuscript. Furthermore, we have carefully checked and fixed any errors in the paper to make sure it follows PLOS ONE's style requirements.

Additional Editor Comments to the Authors:

1) The abstract (under objective) should define participation. Exactly, participation in what? 

Response: Thanks for the advice. We have rewritten the abstract objective as “This study aimed to assess changes in social contact with family, friends and healthcare providers, as well as social participation in working, volunteering, religious services and other organized activities, among older adults during the COVID-19 pandemic while examining the role of pre-COVID sociodemographic characteristics or cognitive and physical limitations in changes in social contact and participation” (Page 3). 

Meanwhile, we polished the first research aim in our introduction as “The goals of this study were to investigate: 1) social contact with family, friends and healthcare providers, social participation in working, volunteering, religious services and other organized activities, and their changes during the COVID-19 pandemic among US older adults” (Page 6).

2) Please clarify informed consent on page 8. Does the manuscript now explain that the survey administrators did not collect consent?

Response: Thank you for pointing this out. In our ethics statement on page 7, we have stated that “The National Health and Aging Trends Study (NHATS) is sponsored by the National Institute on Aging and is being conducted by John Hopkins University Bloomberg School of Public Health. John Hopkins Bloomberg School of Public Health Institutional Review Board approved this study, and written informed consent was obtained from all the involved participants or their proxy respondents. As a secondary analysis using NHATS data in the public domain and anonymized, this research did not fall within the regulatory definition of research involving human subjects and thus was exempt from institutional board review. As a compatible use with what the participants have agreed to in the original consent form, we did not collect another one.

”

3) In describing the regression approach, clarify if/that standard errors were adjusted to correct for heteroscedasticity.

Response: Thank you for this suggestion. We have added the suggested content to our Statistical Analyses paragraph on Page 11 as “To correct for heteroscedasticity, we utilized multivariate linear regression models with Huber–White robust standard errors to separately investigate associations …” The regression estimates with Huber–White robust standard errors were presented in Table 2 and Table 3 with updated notes.

4) Please respond more fully to reviewer comment 15, re: acknowledging the impact of the lack of representativeness in the sample impact on your ability to generalize results. I believe the current claim oversells. This discussion in the manuscript should include other demographic characteristics (such as race) in addition to education. If the you wish to claim more generalizability, the manuscript needs to utilize and engage with weighting estimates at a minimum.

Response: We agree that this is a potential limitation of the study, which has been added and polished in our limitation part “Additionally, most participants reported non-Hispanic White race and were relatively well-educated, making them more likely to have access to technology within their home. Thus, the reported experience and challenges of our participants during the COVID-19 period may not represent the experience of older adults who lacked such facile access, which limits the generalizability of the findings.”

---

## [Editor Report · Decision Letter 2]

12 Apr 2023

Crisis Response During the COVID-19 Pandemic: Changes in Social Contact and Social Participation of Older Americans

PONE-D-22-10689R2

Dear Dr. Clouston,

We’re pleased to inform you that your manuscript has been judged scientifically suitable for publication and will be formally accepted for publication once it meets all outstanding technical requirements.

Kind regards,

Michael E Martell

Academic Editor

PLOS ONE

Additional Editor Comments (optional):

In the copyediting process, you may want to add an `s' to John for Johns Hopkins University.
---

## [Editor Report · Acceptance letter]

20 Apr 2023

PONE-D-22-10689R2 

Crisis Response During the COVID-19 Pandemic: Changes in Social Contact and Social Participation of Older Americans. 

Dear Dr. Clouston:

I'm pleased to inform you that your manuscript has been deemed suitable for publication in PLOS ONE. Congratulations! Your manuscript is now with our production department. 

Kind regards, 

on behalf of

Dr. Michael E Martell 

Academic Editor

PLOS ONE